# A Clinical Picture of Unselected Patients with Systemic Lupus Erythematosus in a Tertiary Hungarian Center—A Spectrum Ranging from Pure Lupus to Overlap Syndromes

**DOI:** 10.3390/jcm13113251

**Published:** 2024-05-31

**Authors:** Dalma Loretta Csóka, Katalin T. Kovács, Gábor Kumánovics

**Affiliations:** Department of Rheumatology and Immunology, Medical School, University of Pécs, 7632 Pécs, Hungary; csoka.dalma@pte.hu (D.L.C.); tkovacs.katalin@pte.hu (K.T.K.)

**Keywords:** systemic lupus erythematosus, classification criteria, sensitivity, autoantibodies, overlap syndrome, interstitial lung disease, EULAR/ACR

## Abstract

**Introduction**: Systemic lupus erythematosus (SLE) is a multidimensional disease; however, the association of another systemic autoimmune disease further complicates its clinical presentation. **Aim**: We decided to investigate whether the association of overlap syndromes is linked with a different clinical picture compared to pure lupus and whether this association changes the sensitivity of the following commonly used criteria: the 2019 European League Against Rheumatism/American College of Rheumatology (EULAR/ACR), the ACR-1997 and the 2012 Systemic Lupus International Collaborating Clinics (SLICC) criteria. **Method**: We performed a retrospective observational study among 382 patients afflicted with lupus: we measured as much of the full clinical and laboratory picture as possible in an unselected cohort. The diagnosis of SLE and other systemic autoimmune diseases was established by the rheumatologist in routine care and then the authors compared the characteristics of patients with pure lupus and those with overlapping pathologies. The diagnosis rates were compared to those that were determined based on the three classification criteria in order to identify various sensitivities and whether the existence of an overlap affects their rates. The fulfillment of each set of criteria was calculated using an Excel-based automatic calculation. **Results**: Among the patients, the ACR 1997′s sensitivity was 81.2% (310 patients), and the SLICC 2012 criteria achieved 94.5% sensitivity (361 patients). The 2019 EULAR/ACR classification criteria resulted in a slightly lower sensitivity (90.3%—345 patients) when compared to the original publication (96%) due to the lower sensitivity of our anti-nuclear antibody (ANA) test (measured via enzyme-linked immunosorbent assay (ELISA)). Nearly all ANA-negative (21/22—95%) patients showed a positive lupus-associated antibody test. The proportion of ANA-negative cases showed no significant difference among pure and overlap patients. No significant difference was found between patients with overlap (138 patients—36%) and pure SLE (244 patients—64%) through the use of these criteria, with the exception of the SLICC criteria (ACR: 80.4% vs. 81.6%; SLICC: 97.4% vs. 92.6%, *p* = 0.035; EULAR/ACR 2019: 91.4% vs. 89.6%). Patients with an overlap syndrome were significantly older (55 vs. 50 years, *p* = 0.001), more likely to suffer from interstitial lung disease (ILD: 20% vs. 11%, *p* = 0.0343) and less frequently showed class III/IV lupus nephritis (7% vs. 14%, *p* = 0.029) when compared with their pure lupus counterparts. **Conclusion**: All investigated criteria regarding sensitivity were similar to the original publication’s findings. The sensitivity of the EULAR/ACR 2019 classification criterion in cases with overlap syndrome proved excellent, with results very similar to patients afflicted with pure SLE. In the presence of an overlap syndrome, we found significantly fewer patients with lupus nephritis III/IV but no differences in other typical lupus organ manifestation beyond the kidney, whereas we found a higher proportion of ILD in patients with an overlap, indicating that the presence of an overlap syndrome significantly influences the observed clinical picture in real-world conditions.

## 1. Introduction

Systemic lupus erythematosus (SLE) is an autoimmune disease affecting several organs. Patients present with a wide range of symptoms causing from mild to life-threatening conditions. This is further complicated by the fact that some symptoms do not necessarily appear at the same time. In terms of differential diagnosis, it is important to highlight that, in addition to lupus, infection, neoplasia, metabolic diseases, medication side effects, and even overlap syndromes are oftentimes lurking behind the symptoms [1].

It should be emphasized that an appropriate and sufficiently reliable classification in the early stages of the disease enables both the early initiation of effective treatment and the inclusion of SLE patients in clinical trials on time [2]. This makes it possible to prevent serious complications and irreversible organ manifestations by intervening during the early stages of the disease, thereby ensuring a better quality of life for our patients [3].

One of the longest-used SLE classification criteria was established by the American College of Rheumatology (ACR) [4]. Alternative methods for SLE classification have been developed, predominantly for adults, such as the SLICC (Systemic Lupus International Collaborating Clinics) criteria and the EULAR/ACR criteria [2,5].

The EULAR/ACR criteria were tested, simplified, and validated in a large (*n*  =  2271) international cohort. Performance characteristics unveiled a sensitivity similar to the SLICC criteria (98% versus 95% for the SLICC and 85% for ACR 1997) while maintaining the specificity of the ACR 1997 criteria (97% versus 95% for the ACR and 90% for the SLICC) [2]. Additionally, it is important to note that when compared to the previous criteria, the new system of criteria introduced two new conceptions: ANA testing as an obligatory entry criterion and the weighing of an individual criterion.

It is generally accepted that autoimmunity is essentially a spectrum where differentiation is made for didactic and therapeutical purposes on the basis of a characteristic combination of predominant organ involvements. This also means that in everyday life, we often encounter atypical cases, including those in which multiple systemic autoimmune conditions can be considered to co-exist. However, there is a very limited amount of literature that has thoroughly investigated the impact of an associated systemic autoimmune disease regarding the observed clinical picture, including either the most clinically significant and less frequently observed manifestations, as well as its possible implications for the applicability of the classification criteria. A relatively large number of publications have examined the performance of each set of criteria in relation to geographical and ethnic backgrounds [6]. There was also one that looked at the same criteria in a narrower subgroup, e.g., SLE with a long disease duration [7]. One publication investigated the effect of revising criteria in different connective tissue diseases (CTDs), in which the newer criteria showed increasing sensitivity [8]. On the other hand, in the last few years, only a small number of publications have analyzed lupus overlap disease, and none of them have examined the impact of the presence of an overlap disease on the fulfillment of the criteria [9,10,11]. Furthermore, the literature was not clear on the prevalence of such important organ manifestations as lupus nephritis in patients with overlap syndrome [9,10].

## 2. Objective

The main aim of our study was to assess the impact of an association of connective tissue diseases with SLE regarding its full spectrum of clinical and laboratory characteristics using data from a single center, including manifestations that are not part of the classification criteria but are relevant to the clinical picture of the disease. We decided to determine whether the presence of overlap diseases modifies the frequency of major organ manifestations, i.e., those with the most important prognostic characteristics (without assessing their severity). Another key objective of our study was to investigate whether or not the presence of overlap syndromes alters the sensitivity of the three most commonly used criteria systems.

## 3. Methods

The study was conducted according to the guidelines of the Declaration of Helsinki and approved by the Institutional Review Board (Data Protection and Research Ethics Committee, University of Pécs, Hungary, code: KK/971-1/2020, date of approval: to 25 September 2020).

The data collection occurred based on the query of MedSolution (the electronic medical record system of our center) reports using the following SLE-BNO codes (the Hungarian version of the International Classification of Diseases)—BNO: 3210, 3280, and 3290. Screening patients included cases in which there were at least 2 appearances. We collected the patient’s sex, age, associated diagnoses, the duration of their illness, follow-up time, comorbidities, symptoms associated with various organ manifestations, as well as other clinically important symptoms not included in the criteria systems.

We accepted the existence of SLE in a patient if it was definitively described in the medical opinion, with a supporting list of clinical and laboratory data in medical records on multiple occasions during the follow-up. However, the diagnosis had not been confirmed by an external independent auditor. The fulfillment of each set of criteria was calculated using an Excel-based automatic calculation, with each manually controlled by the first author.

With regard to overlap syndromes, the clinical diagnosis was made based on the treating physician’s medical opinion and based on principles similar to those used to establish the diagnosis of lupus. On at least two occasions, we found a definitive opinion about this diagnosis in the medical records, together with a list of supporting arguments. Furthermore, the following findings are uniformly true: systemic sclerosis (SSc) was considered when either SSc-specific antibody positivity or a clear scleroderma pattern was detected by capillary microscopy in addition to sclerodactyly. Myositis was diagnosed if an elevated creatine kinase value was found in the presence of muscle weakness and either a positive muscle biopsy or myositis-specific antibody positivity. Rheumatoid arthritis (RA) was defined when high titers (>3×) of rheumatoid factor (RF) or anti-cyclic citrullinated peptide (aCCP) positivity were detected in addition to polyarthritis. Sjögren’s syndrome was considered if the patient suffered either objective sicca abnormalities with anti-Sjögren’s syndrome antigen A and B (SSA/SSB) positivity or positive lip biopsy results (focal lymphocytic sialoadenitis: focus score ≥1 per 4 mm^2^) with sicca complaints. Antiphospholipid syndrome (APS) was evaluated if the patient experienced either deep vein thrombosis, a pulmonary embolism, or multiple miscarriages in addition to specific laboratory tests (anti-cardiolipin (aCL), anti-β2-glycoprotein I (aB2GPI) antibody, and lupus anticoagulant (LA) positivity).

The diagnosis established, based on the opinion of the attending physician, was next compared with the above three diagnostic criteria systems, with a diagnosis ratio that can be set up based on a classification criteria system, determining the sensitivity of each criteria system.

## 4. Results

The diagnosis of SLE was established in 382 patients. Out of 382 patients, 345 (90.3%) were female. The average age at onset was 34 ± 14 years, while the average disease duration was 17 ± 10 years. We established the existence of a further diagnosis of another connective tissue disorder in 138 (36%) cases.

To quantify the serological items, anti-nuclear antibody positivity (ANA) was detected in 94% of the patients tested via enzyme-linked immunosorbent assay (ELISA). From 22 ANA-negatives 21/22, 95% of patients showed a positive lupus-associated antibody test: we established 17 patients with an anti-double-stranded DNA antibody (dsDNA), two patients with antiphospholipid, one patient with SSA, and another one with C1q positivity. Anti-dsDNA positivity was observed in 332/382 (87%) cases. Anti-Sm was observed in 45/382 (11.8%) cases. Antiphospholipid antibodies (aPLA) were found in 64% of patients (245/382 patients). Anti-CL experienced a prevalence of 47% (180/382), aB2GPI positivity occurred in 39% of patients (150/382), and positive LA occurred in 32% of the patients (124/382). Double positivity (aCL and aB2GLPI) manifested in 122/382 (32%) cases and 58/382 (15%) patients experienced triple positivity.

In the evaluation of the sensitivity regarding the three classification criteria, the ACR 1997 and SLICC 2012 criteria achieved similar sensitivity to that in the original publication: 81% versus 83% for ACR 1997 and 95% versus 97% for SLICC 2012. In comparison, the newest 2019 EULAR/ACR classification criteria experienced a sensitivity of 90% versus 96% (Figure 1). However, adding dsDNA testing as an entry criterion strengthened the sensitivity to 95%.

The data show a higher prevalence in neurologic involvements investigated by the SLICC criteria when compared to the 2019 EULAR/ACR criteria, 78/361 patients (21.6%) vs. 33/345 patients (9.6%), with a *p* < 0.001, which was independent of the addition of dsDNA results to ANA positivity. The main reason may likely be that the SLICC criteria contain more neurologic conditions, such as mononeuritis multiplex (0.8%), myelitis (1%), cranial (2.1%), peripheral neuropathy (9.9%), and acute confusional state (1.3%), affecting higher prevalence, which is not included in the newest criteria. By using the ACR 1997 criteria, 32/310 (10.3%) cases had neurologic manifestations (seizure, psychosis, or both) (Table 1).

However, renal involvement had almost the same occurrence when examined using the SLICC criteria’s 103/361 (28.5%) patients when compared with the 2019 EULAR/ACR criteria’s 94/345 (27.2%) and the ACR 1997 criteria’s 100/310 (32.3%) patients. Additionally, when adding a dsDNA test as an entry criterion to the 2019 EULAR/ACR, a similar prevalence was found (102/361 patients (28.2%)). Proteinuria, as a commonly observed sign in lupus nephritis (LN), was found in 96/105 of the patients who suffer from renal involvement. However, hematuria and cylindruria are not defined separately in the newest criteria. The 2019 EULAR/ACR criteria had the lowest prevalence because red blood cell (RBC) casts or any biopsy-proven LN were not defined in the newest criteria (Table 2).

The prevalence of joint involvement has been similarly found by the ACR 1997 criteria in 281/310 (90.6%) patients, the SLICC criteria in 318/361 patients (87.8%), and the 2019 EULAR/ACR in 303/345 patients (87.8%). Of those individuals who experienced joint involvement, 48% had RF (159/331), 6.3% had CCP (21/331) positivity, and 4.8% of patients (16/331) were reported to be positive with both. Pulmonary fibrosis was assumed in 83/382 (21.7%) cases, which was confirmed via a computed tomography (CT) scan in 55 cases (14.4%). The prevalence of sicca symptoms was 43.7% for all patients in our study (167/382 patients). Looking at the different sets of criteria of SLE, we found lower prevalence rates: by using the ACR-1997, it was detected in 136/310 (43.8%) patients. By following SLICC-2012 criteria, it was detected in 157/361 (43.4%) patients, and the 2019 EULAR/ACR, it was detected in 151/345 (43.7%) patients.

The observed minimal differences in the incidence of mucocutaneous symptoms are also due to differences in definitions (Table 3). In terms of serosal involvement, the three criteria yielded nearly identical results: the ACR 1997 reported 93/310 (30%) patients, the SLICC 2012 reported 99/361 (27.42%) patients, and the 2019 EULAR/ACR reported 97/343 (38.27%) patients, compared to the total of 100/382 (26.2%) patients with serositis.

In considering overlap syndrome, 138 (36%) of the 382 patients with SLE experienced at least two diagnoses of a systemic autoimmune disorder based on the opinion of the treating physician: six patients with SSc, 63 patients with Sjögren’s syndrome, 63 patients with APS, two with idiopathic inflammatory myopathy, 23 with RA, and five patients with vasculitis (three with ANCA-associated vasculitis, one with Henoch–Schönlein Purpura, one with unclassifiable vasculitis) (Table 4). Among these patients, 24 were diagnosed with multiple overlap syndrome. In relation to organ-specific autoimmunity, we found 13 patients with inflammatory bowel diseases, nine patients with autoimmune hepatitis, and 20 patients with thyroid dysfunction. While the sensitivity measured by the ACR and the EULAR/ACR 2019 criteria showed no difference between patients with overlap syndrome and non-overlap SLE (ACR: 80.4% vs. 81.6%; EULAR/ACR 2019: 91.4% vs. 89.6%), the sensitivity was significantly higher in the SLCICC system regarding patients with an overlap (97.4% vs. 92.6%, *p* = 0.035) (Figure 1).

Our patients experiencing an overlap were significantly older compared to those with pure lupus (55 ± 14 vs. 50 ± 14 years, *p* = 0.001). Additionally, their SLE typically manifests later in life. The clinical difference between the two groups was a higher incidence of pulmonary fibrosis (29% vs. 18%, *p* = 0.0138) and sicca complaints (64% vs. 32%, *p* < 0.001) at the expense of those with an overlap. In regard to CT-confirmed ILD, the significant difference between the two groups remained (pure SLE vs. overlap: 11% vs. 20%, *p* = 0.0343). Of the 27 CT-confirmed ILD patients with overlap syndrome, three suffered from SSc, nine suffered from Sjögren’s syndrome, three suffered from RA, one suffered from vasculitis, six suffered from antiphospholipid syndrome, and five suffered from multiple overlap syndrome. In terms of tendency, this group also included more patients afflicted with Raynaud’s phenomenon and fewer cases with renal involvement without reaching a level of significant difference (Table 4). On the other hand, when we analyzed the proportions of subgroups of our nephritic patients, there were significantly fewer patients with class III/IV lupus nephritis among our cases with overlap syndrome compared to pure lupus patients (7% vs. 14%, *p* = 0.029). With regard to laboratory parameters, the anti-C1q antibody was more frequent among patients with pure SLE (27% vs. 17%, *p* = 0.0391), while anti-SSA, LA, and RF were more frequent among patients afflicted with an overlap (Table 4).

## 5. Discussion

The main result of our study is that the association of an overlap syndrome changes the typical clinical picture of SLE, the prevalence of certain organ manifestations may decrease (class III/IV lupus nephritis: 14% in pure SLE vs. 7% in SLE overlaps) or increase (ILD: 11% in pure SLE vs. 20% in SLE overlaps), which may partly influence the sensitivity of the classification criteria (SLICC: 92% in pure SLE vs. 97% in SLE overlaps) and may modify our clinical thinking; e.g., it may be worthwhile to introduce more accurate tests (HRCT: true ILD was only identified in 66% of our cases with positive chest X-rays for pulmonary fibrosis and it is accepted that the sensitivity of a chest X-ray is up to 80% for identifying ILD [12]) as a screening tool in patients with an overlap or to recommend more frequent non-invasive tests (respiratory function).

It is important to emphasize that a set of classification criteria also provides a kind of orienting guide for everyday clinical work. The opposite is also true: when reading and interpreting a publication, it is important to know which elements are emphasized more and which are not in a set of criteria. The present cross-sectional study demonstrated that the differences in the sensitivity of the ACR 1997 and SLICC 2012 criteria are nearly negligible when compared to the original publication’s findings, in contrast with the newest 2019 EULAR/ACR classification criteria with a sensitivity of 90% versus 96%, which seemingly is attributed to the compulsory ANA positivity. The concept of an ANA test being used as an entry criterion was challenged based on data with a higher-than-usual rate of ANA-negative SLE and on technical issues regarding ANA sensitivity [13]. A previous study emphasizing the impact of different methods in autoimmune serology found that 88% of SLE patients experienced a positive ANA test when investigated with immunofluorescence in contrast with different ELISA kits—with results ranging from 62 to 90% [14]. In consideration of an appropriate ANA assay and a correct dilution of serum, the prevalence of the true ANA-negative cases is very low: previous studies showed a 2–6% total prevalence of ANA-negative lupus patients [15,16]. If we do not have an accurate method, we need to find another solution. Our results, demonstrating that adding dsDNA testing as an entry criterion strengthened the sensitivity to 95%, are a potential solution. It is crucial to take into account the addition of this assessment to the testing algorithms commonly used in routine care, which can enhance both the sensitivity of this criteria and the possibility of false-positive results [17]. Therefore, this can be evadable by using a parallel investigation for a dsDNA test or repeating the ANA test with another kit or method if standard indirect immunofluorescence staining cannot be carried out. Our results, in general, with simultaneous dsDNA testing, are identical to those of a recent meta-analysis referencing the diagnostic accuracy of the 2019 EULAR/ACR criteria with a mean sensitivity of 95% [6]. However, despite the similar results, it is important to note one of the limiting features of our present study: that our reference standard diagnosis has not been confirmed by an external independent reviewer.

In addition to the different screening methods regarding ANA, there are several points bearing worthwhile discussion in regard to potential misclassification. The investigation of serum complement levels has become an integral part of the SLICC 2012 and the 2019 EULAR/ACR criteria. We found a low level of serum complement in 168 patients (43.9%). However, there was a lack of suitable documentation in some of our cases; in those that have not been separated, complement fractions have been involved.

With regard to the clinical picture, it is important to emphasize that our study aimed to assess the possible variability in the frequency of organ involvement in a non-selected group of lupus patients, but in addition to the frequency of the observed manifestations, it is also important to assess their severity and prognosis to obtain a complete picture, which we do not believe is possible in a cross-sectional study.

Evaluating the presence of hematological disorders as investigated by the ACR 1997 criteria, 213/310 cases were found (68.7%) compared to the SLICC 2012′s 292/361 (80.8%) patients vs. the 2019 EULAR/ACR’s 212/343 (61.8%) patients. The reason for the higher prevalence when using the SLICC 2012 criteria may be that the criteria contain hemolytic anemia, leucopenia, or lymphopenia at least once in one instance and thrombocytopenia at least once when compared to the ACR 1997, in which it is required to have two or more occurrences of these, and in the newest criteria lymphopenia is not defined separately. Overall, our data regarding hematological symptoms are similar to previously published findings [18,19,20,21].

Varying degrees of renal involvement can be found in SLE patients, manifesting as glomerular, tubular, interstitial, or vascular changes [22]. Our own data show a slightly lower prevalence of lupus nephritis (27%) when compared to results published in the previous literature (40–50%) [23] (milder patients are also cared for in our clinic), yet no significant difference was found for each criterion (Table 2). However, in some recent single-center studies of similar size, the proportion of patients with lupus nephritis was lower than that in our study: 103 (19.2%) of 536 SLE patients in a Saudi study were clinically diagnosed with lupus nephritis [24]. In another single-center study from Germany, 122 (22.4%) out of 545 patients were biopsy-confirmed as having class II/III/IV/V lupus nephritis [25]. In a British study, although 33% of all patients (n = 496) were diagnosed with nephritis, the rate in the Caucasian ethnic group was much lower at only 25% [26]. However, we do not have specific data on ethnicity for our own patients.

Neuropsychiatric (NP) features are one of the main organ involvements among SLE patients with various clinical presentations. However, not all symptoms are associated with lupus disease activity, which makes it difficult to clearly distinguish the NP events due to SLE from those attributed to non-SLE causes. The reported frequency of NP lupus shows a wide range in the published literature [27]. In perusing through the investigated criteria, the range of neurologic manifestations was expanded in the SLICC 2012 criteria to contain more unspecific conditions (like mononeuritis multiplex (0.8%), myelitis (1%), cranial (2.1%) and peripheral neuropathy (9.9%)), which were reduced for the 2019 EULAR/ACR criteria due to playing a minor role in classification according to a review conducted by Martin Aringer et al. [28] (Table 1).

In consideration of serosal involvement, nearly the same occurrence was found when investigated by the three criteria: with the ACR 1997, 93/310 patients (30%) were found vs. SLICC 2012, which found 99/361 patients (27.4%), and the 2019 EULAR/ACR, which found 97/343 (38.3%), when comparing all 100/382 (26.2%) patients with serositis (Table 3). A study conducted by Yan Liang et al. reported a 17.9% possible prevalence of serositis, which was identified if any signs and/or symptoms of pleuritis and/or pericarditis were presented [29]. The presence of ascites was also observed in 8.1% of patients (31/382 patients). Nevertheless, there was no mention of peritoneal serositis in either of the three investigated criteria. In some of our cases, the exact meaning was not properly defined in reference to serositis, including pericarditis, pleuritis, and peritonitis, and was not separated, whether it was an acute or a chronic serosal complication. This highlights the need to accurately document all clinical and laboratory results in routine patient care, both for the data included and not included in the criteria, as the criteria may change over time.

SLE-related arthritis is a common clinical manifestation occurring in approximately 90% of patients, typically presenting as non-erosive arthritis, which closely correlates with our findings—it was present in 331/382 patients (86.6%). However, as previously illustrated, it was not defined as the primary feature of arthritis, regardless of whether any imaging techniques considered it a non-erosive arthritis or an RA-like erosive pattern. However, neither the evidence of erosive phenotypes nor the positivity of certain immuno-serological tests (RF, CCP) provide clear help in differentiating arthritis in the two diseases; thus, the true SLE-RA overlap syndrome, Rhupus, remains a current diagnostic problem [9,30].

Moreover, several clinical features were also gathered that are not part of the three classification criteria due to their lack of specificity; however, this can prove to be clinically significant. We noticed a high prevalence of Raynaud’s phenomenon (214/382 patients (56%)), lymphadenopathy (83/382 patients (21.7%)) and hepatosplenomegaly (120/382 patients (31.4%)). Pulmonary fibrosis, which is not included in the criteria systems, however, is not merely therapeutic but bears prognostic significance, too, since the possible presence of it arose in a large number of patients (83/382 patients (21.7%)). However, it was confirmed in only 55 cases (14%) using high-resolution computed tomographic scans, which is very similar to previous findings with a prevalence of 13–17% [31]. All these clinically significant features can improve the sensitivity of new classification criteria at the cost of having lower specificity.

In addition to investigating the presence of the most common comorbidities, the existence of multiple connective tissue diseases was also observed in 36% of the study’s population. APS and Sjögren (16–16%) were the most commonly found overlap syndromes among our patients. It is also essential to separate the SLE-APS overlaps from the aPLA-positive cases due to their different mortality rates and therapeutic consequences. APL positivity experienced a high prevalence (245 patients—64.1%) in our patients, which is higher when compared with the previously published literature: the accepted rate of antibody positivity is 30–50%, while the prevalence of forms with clinical events similar to ours was between 10 and 20% [32,33]. Multiple overlaps associated with SLE were found in 6% of patients. According to our results, there is no significant variance between overlap vs. non-overlap patients with the oldest and the latest criteria. In contrast, for the SLICC criteria, we found significantly higher sensitivity in patients with overlap (Figure 1), suggesting the lower specificity of this criterion in overlap syndrome; however, we cannot confirm this in the absence of specificity studies. Nevertheless, the existence of overlap syndromes was established based on the clinicians’ opinions and it was not investigated whether it fulfilled the classification criteria or not, which is another limitation to consider. In a recently published paper with 21% of the lupus patients studied having overlap syndromes, the authors found a very similar sensitivity, which was 82.9% for the 1997 ACR criteria and 92.4% for the 2019 EULAR/ACR criteria [8]. Overall, in SLE with overlap syndrome, we obtained results with a sensitivity comparable to the original publications, which was at the same time similar to the few results in the published literature to date in this area.

Relatively few previous publications thus far have examined the association between connective tissue diseases within a single SLE cohort [9]. Most frequent association rates are very similar for both Sjögren’s and APS, and these data are confirmed by our own results (16–16%). Similarly, it is also known that the age of lupus patients with overlap cases is higher than that of those with pure SLE (55 vs. 50-year-olds). Several data in the published literature suggest renal manifestations are less frequent in overlap cases [9]; however, this was only a trend in our cohort (22% vs. 31%), similar to other previous observations [10]. However, when looking, in detail, at renal involvement, there were significantly fewer patients with III/IV lupus nephritis than there were patients with pure lupus (7% vs. 14%). To the best of our knowledge, this result has not yet been investigated or reported by another working group, so confirmation in another cohort with a larger number of patients is still needed. In terms of trends, a similar decrease (not reaching significance) was observed in class II/V lupus nephritis (for patient with pure lupus: 8% vs. 4% in those with overlap), but no difference was found in the classical clinical indicators such as the proteinuria, hematuria, and cylindruria rates between patients with overlap and pure lupus. This further strengthens our view that in the presence of overlap syndrome, the pursuit of harder endpoints is recommended, making the performance of, e.g., renal biopsy even more justified (to exclude a mild or even non-lupus origin, to assess associated vasculitis). Neurologically, we found no significant difference between the two groups. We know from a previous study that peripheral neuropathy is more common in lupus patients with secondary Sjögren’s syndrome compared to those with pure lupus [11], but in our cohort this was not the case for the whole overlap subgroup (11% vs 9% in pure lupus). As in this previous publication [11] we also observed a higher rate of pulmonary fibrosis among patients with overlap (20% vs 11% in pure lupus) (Table 4). This latter finding is supported by another report in which both the associations with other connective tissue diseases (CTDs) and older age were associated with a higher incidence of ILD [34]. A chest X-ray provides only limited specificity and sensitivity and is primarily used to rule out other diseases. However, it is well known that ILD is initially clinically asymptomatic and that in at least one-third of patients with clinically definite CTD, the HRCT examination supports the presence of ILD without other clinical signs [31]. This also means that we will not be able to diagnose an asymptomatic ILD in a significant proportion of our patients unless we routinely perform HRCT scans, at least in a patient population where the chances of this are significant: our overlap patients showed ILD at nearly twice the frequency of our purely SLE patients. However, the results of our current study do not allow us to assess the long-term prognosis of the altered clinical picture of lupus associated with overlap syndrome, and therefore, prospective studies are needed.

## 6. Conclusions

In analyzing the various criteria, the main differences in the performances consist of changes in the definition of organ involvement, the score of items, and the additive weighted scoring system. According to our data, the sensitivities were similar to those of the original publication’s findings, yet in some patients, our ANA ELISA test showed false-negative results. In the case of using another method, such as standard indirect immunofluorescent staining (on HEp-2 or Crithidia luciliae), we recommend the parallel investigation of a dsDNA test and a preparatory analysis of the description of the available ANA test. In the presence of overlap syndrome, we encourage the use of the new EULAR/ACR criteria since they have shown nearly equal yet excellent sensitivity in both overlap and pure lupus cases. When reading the frequency of each organ manifestation in the literature, we should always consider not only which criteria they based their study on but also whether they gave the proportion of the cases with overlap syndrome. The sensitivity of the more permissive SLICC criteria was significantly different in patients with overlap syndrome compared to pure lupus patients. Our results will, therefore, move our thinking towards the use of “harder endpoints” (more important/frequent manifestations with worse prognosis) when making diagnoses in a case where the possibility of multiple CTDs is raised in practice. Patients afflicted with overlap syndrome tend to exhibit less frequent renal involvement, which is significant for class III/IV lupus nephritis, yet significantly more frequent ILD is confirmed via CT. Although dangerous, lupus nephritis may be considered less common, but should still be expected; meanwhile, the rate of lung involvement is increasing, so our attention should not wane in any respect but rather be raised towards other organ involvement, such as ILD. Therefore, in a patient with lupus who has the possibility of other CTDs, we should change the sensitivity of our screening tests (e.g., requesting HRCT instead of a chest CT) and even their frequency (respiratory function tests being run not every few years but rather every 3–6 months), in addition to the need to carry out standard lupus-oriented tests at an unreducible quality and frequency. Furthermore, not only for the lungs but also for the kidneys, our data suggest that a more accurate method of investigation is recommended in the presence of overlap syndrome, and thus, it is even more recommended to perform a renal biopsy to exclude/reject overlap abnormalities, even in the presence of mild clinical signs. A long-term follow-up protocol regarding these patients is necessary to better assess the prognosis of patients suffering from overlap syndrome because we do not yet have sufficient information on the severity and long-term outcome of organ involvement rates.

## Figures and Tables

**Figure 1 jcm-13-03251-f001:**
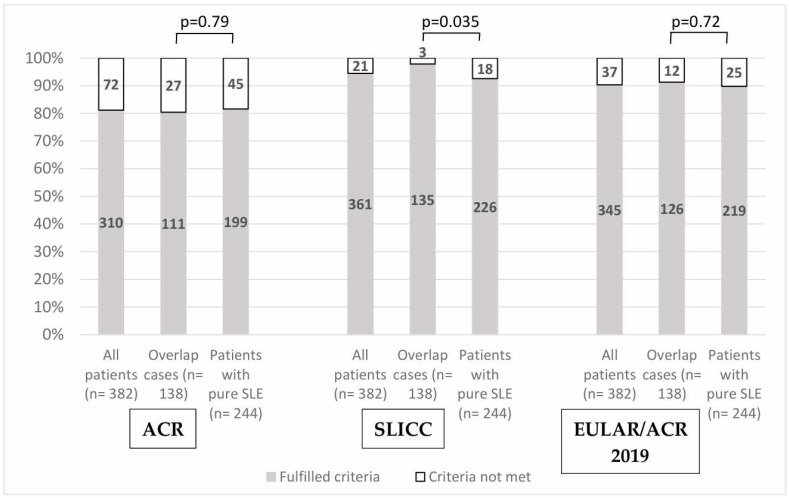
A comparison of the sensitivities measured by the three criteria systems tested in all patients with overlap syndrome and pure SLE. The values in the columns indicate the number of patients affected. The total number of investigated patients with lupus is 382. The number of patients for each criterion: ACR = 310, SLICC: 361, EULAR/ACR 2019: 345. The number of patients with overlaps: 138. The number of patients with pure lupus is 244.

**Table 1 jcm-13-03251-t001:** The prevalence of nervous system involvement in all our patients, according to each criterion.

Symptoms	Cases n (%)	ACR-1997	Cases n (%)	SLICC 2012	Cases n (%)	2019 EULAR/ACR	Cases n (%)
Seizures	23 (6.0)	Seizures	23 (7.4)	Seizures	23 (6.4)	Seizures	23 (6.7)
Psychosis	7 (1.8)	Psychosis	7 (2.3)	Psychosis	7 (1.9)	Psychosis	7 (2.0)
Delirium	5 (1.3)			Acute confusional state	5 (1.4)	Delirium	4 (1.2)
Mononeuritis multiplex	3 (0.8)			Mononeuritis multiplex	3 (0.8)		
Myelitis	4 (1.0)			Myelitis	4 (1.1)		
Cranial neuropathy	8 (2.1)			Cranial neuropathy	8 (2.2)		
Peripheral neuropathy	38 (9.9)			Peripheral neuropathy	36 (9.9)		

The number of patients for each criterion: ACR = 310; SLICC: 361; EULAR/ACR 2019: 345. The total number of investigated patients with lupus is 382.

**Table 2 jcm-13-03251-t002:** The prevalence of renal manifestations in all our patients compared using each criterion.

Symptoms	Cases n (%)	ACR 1997	Cases n (%)	SLICC 2012	Cases n (%)	2019 EULAR/ACR	Cases n (%)
Proteinuria > 0.5 g/24 h	96 (25.1)	Persistent proteinuria	92 (29.7)	Urine protein/creatinine or 24 h urine protein > 0.5 g	95 (26.2)	Proteinuria > 0.5 g/24 h	87 (25.2)
Cylindruria	16 (4.2)	Cellular casts	85 (27.4)	RBC	72 (19.9)	WHO II/V LN	23 (6.6)
RBC	73 (19.1)			Biopsy-proven LN and positive ANA or anti-dsDNA	72 (19.8)	WHO III/IV LN	43 (12.5)
WHO II/V LN	27 (7.1)						
WHO III/IV LN	46 (12.0)						
All biopsy-proven LN	73 (19.1)						
Clinically diagnosed LN	105 (27.5)						

LN: lupus nephritis. RBCs: red blood cell casts. The number of patients for each criterion: ACR = 310; SLICC: 361; EULAR/ACR 2019: 345. The total number of investigated patients with lupus is 382.

**Table 3 jcm-13-03251-t003:** The difference in the prevalence of mucocutaneous involvement among different criteria.

Symptoms	Cases n (%)	ACR 1997	Cases n (%)	SLICC 2012	Cases n (%)	2019 EULAR/ACR	Cases n (%)
Malar rash Discoid rash ACLA PS Oral ulcers Alopecia Chillblain	262 (68.6) 21 (5.5) 11 (2.9) 191 (50.0) 58 (15.2) 148 (38.7) 2 (0.5)	Malar rash Discoid rash PS Oral ulcers	151 (48.7) 19 (6.1) 185 (59.7) 54 (17.4)	ACLE SCLE/DLE Oral ulcers Alopecia	256 (70.9) 20 (5.5) 57 (15.7) 145 (40.1)	ACLE SCLE/DLE Oral ulcers Alopecia	246 (71.3) 20 (5.8) 56 (16.2) 134 (38.8)

PS: photosensitivity; ACLE: acute cutaneous lupus erythematosus; DLE: discoid lupus erythematosus; subacute cutaneous lupus erythematosus. The number of patients for each criterion: ACR = 310, SLICC: 361, EULAR/ACR 2019: 345. The total number of investigated patients with lupus is 382.

**Table 4 jcm-13-03251-t004:** The main clinical and laboratory data of patients with overlap syndromes and pure SLE.

	Patients with Pure SLE n (%)	Patients with Overlap n (%)	Significance
Sex, female	221 (91)	124 (89)	*p* = 0.86
**Age (mean ± SD) years**	**50 ± 14**	**55 ± 14**	***p* = 0.001**
Age at diagnosis (mean ± SD) years	34± 14	37 ± 15	*p* = 0.073
Disease duration (mean ± SD) years	16 ± 10	18 ± 10	*p* = 0.10
Hypertonia	119 (49)	74 (54)	*p* = 0.39
Diabetes mellitus	22 (9)	17 (12)	*p* = 0.38
Smoking	64 (26)	35 (25)	*p* = 0.90
Malignancies	23 (9)	16 (12)	*p* = 0.49
Raynaud’s syndrome	128 (52)	86 (62)	*p* = 0.06
**Sicca symptom**	**78 (32)**	**89 (64)**	***p* < 0.001**
Fever	70 (27)	36 (26)	*p* = 0.64
Lymphadenopathy	54 (22)	29 (21)	*p* = 0.90
Photosensitivity	119 (49)	72 (52)	*p* = 0.59
Hepatosplenomegaly	78 (32)	42 (30)	*p* = 0.82
**Pulmonary fibrosis on chest X-ray**	**43 (18)**	**40 (29)**	***p* = 0.014**
**ILD with HRCT**	**28 (11)**	**27 (20)**	***p* = 0.034**
Thyroid disorder	14 (6)	6 (4)	*p* = 0.64
Joint involvement	213 (87)	118 (86)	*p* = 0.64
Hematological abnormality	138 (57)	90 (65)	*p* = 0.10
Mucocutaneous signs	204 (84)	120 (87)	*p* = 0.46
Neuropsychiatric lupus	46 (19)	35 (25)	*p* = 0.15
Renal manifestation	75 (31)	30 (22)	*p* = 0.073
Proteinuria > 0.5 g/24 h	67 (27)	29 (21)	*p* = 0.18
Cylindruria	11 (5)	5 (4)	*p* = 0.79
Hematuria	52 (21)	21 (15)	*p* = 0.18
Class II or V lupus nephritis	20 (8)	6 (4)	*p* = 0.20
**Class III or IV lupus nephritis**	**35 (14)**	**9 (7)**	***p* = 0.029**
Anti-nuclear antibody presence	227 (93)	133 (96)	*p* = 0.25
Anti-dsDNA presence	218 (89)	116 (84)	*p* = 0.15
Anti-SM presence	29 (12)	16 (12)	*p* = 1.00
Anti-cardiolipin presence	107 (44)	74 (54)	*p* = 0.07
Anti-β2 glycoprotein presence	91 (37)	60 (43)	*p* = 0.28
**Lupus anticoagulant** presence	**61 (25)**	**64 (46)**	***p* < 0.001**
**Anti-C1q** presence	**58 (27)**	**24 (17)**	***p* = 0.039**
**Anti-SSA** presence	**80 (33)**	**70 (51)**	***p* < 0.001**
Anti-nucleosome presence	166 (68)	97 (70)	*p* = 0.73
**Rheumatoid factor presence**	**83 (34)**	**76 (55)**	***p* < 0.001**
Anti-CCP presence	10 (4)	12 (9)	*p* = 0.071
Direct Coombs positivity	33 (14)	26 (19)	*p* = 0.19
Low complement	107 (44)	61 (44)	*p* = 1.00

The number of patients with pure lupus is 244. The number of patients with an overlap is 138. ILD: interstitial lung disease. HRCT: high-resolution computer tomography. Renal manifestation: based on the clinician’s opinion. Hematuria: >3 red blood cells in urine sediment/high power field (HPF). Cylindruria: >0 cylinders in urine sediment/HPF. Low complement: below the normal reference range in our central laboratory—for C3: 0.90–1.80 g/L, for C4: 0.10–0.40 g/L.

## Data Availability

The data presented in this study are available on request from the corresponding author.

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
