# Peer review of "A Clinical Picture of Unselected Patients with Systemic Lupus Erythematosus in a Tertiary Hungarian Center—A Spectrum Ranging from Pure Lupus to Overlap Syndromes"

_jcm, 2024, doi:10.3390/jcm13113251_

Round 1

Reviewer 1 Report

Comments and Suggestions for Authors

A potentially interesting article on SLE and classification criteria, however, I have some comments that have to be addressed before the final decision.

First of all, a clear aim of the study should be stated in the abstract and manuscript regarding the novelty of the data.

Please change gender into sex.

Please write a full form of abbreviated words when they appear for the first time (SSc, RA, etc.).

Please write the exact p-values.

Line 182. For older or younger, please see 50 vs. 55 years.

Statistically significant p-values should be presented rounded to the third decimal place (in the case of minimal values, p<0.001), and those with a value > 0.05 to the second decimal place.

Tables should be n (%).

Please focus in the first paragraph in the discussion section on the novelty of the paper or what kind of new information this research adds to the current knowledge.

Please compare the occurrence of lupus nephritis with other single-center studies in the discussion, that might be more consistent with your single-center study work, especially the newest papers, like from this year.

Lines 256, 260. Please correct et al.

Did you collect a permission form the bioethical committee?

More clinically relevant information should be addressed in the conclusion section to make data more practical.

Please follow the journal's regulations regarding the reference style.

Comments on the Quality of English Language

Moderate editing of English language required. Please recheck the paper.

Author Response

Thank you very much for your many constructive and clarifying comments!

Reviewer 2 Report

Comments and Suggestions for Authors

This retrospective observational study was conducted at a tertiary Hungarian Medical Center, focusing on 382 patients diagnosed with systemic lupus erythematosus (SLE). The study aimed to assess the clinical and laboratory features of patients with pure SLE and those with overlap syndrome, and to evaluate the performance of the 2019 European League Against Rheumatism/American College of Rheumatology (EULAR/ACR) classification criteria compared to earlier criteria (ACR 1997 and SLICC 2012).

There were 3 parts of the study:

1. global performance of the 3 classification criteria (in patients with LES, compared to patients with LES+another=overlap)

2. the frequency and distribution of the associated diseases

3. the frequency of organ involvement. This last aspect was biased because the frequency was calculated in different sample sizes of SLE patients diagnosed using the three classification criteria (310 patients with SLE by ACR criteria, 361 by SLICC, and 345 by EULAR/ACR). However, the existing diagnosis was considered the reference standard. Therefore, we don't see the rationale for calculating the prevalences on different numbers of patients.

The authors should explain the clinical utility of all their objectives and findings and stress the relatively low validity of their reference standard.

Author Response

(The authors gave the same response as above.)

Round 2

Reviewer 1 Report

Comments and Suggestions for Authors

Thank you for the chance to reassess the paper. An article was only partially improved. The way of presentation is below the standards for such prestigious journals like JCM. Please see my comments. I am not sure if the main concept of the work is good and in some cases it is misleading. I believe that the paper need more extensive revisions.

Please include overall number of cases in Tables, like: n (%).

Please add footnotes to each tables with abbreviated forms explained, all, X - (explanation).

Statistically significant p-values should be presented rounded to the third decimal place (in the case of very small values, p<0.001), and those with a value > 0.05 to the second decimal place. Please correct it. See tables. Do not write NS, please add exact p-values.

Figure 1. It is not known if numbers are of all patients, please write the No of all patients below, e.g. All patients n = 310; Overlap cases n = 111

Table 4. Please change into: Sex, female, n (%). Raynaud => Raynaud's syndrome? Smoking? Please be more precise. Immunserology? Antinuclear body presence, Rheumatoid factor presence, n (%).

How was low complement defined?

Abstract. The beginning of the abstract should be more informative and present the background of the paper. Please do not use the form, aim: ... Instead, write, Thus, we decided to investigate ... or similar form.

EU-LAR/ACR? It is EULAR/ACR

Line 18. with SLE

Lines 21-22. How was the sensitivity checked? More methods have to be described.

Do conclusions reflect the aims of the study?! Please see the last sentence.

Please add EULAR/ACR to keywords

Lines 42-47. References?

Lines 64-66. It does not reflect the presented background in the study, aims of the work, and title. Please rethink to be precise in this matter.

Objectives. What actually does this study add to the current literature work?

Rf => RF

Line 97. (aCCP) antibodies

Line 100. 4 mm^2

Line 103. aCL

Line 104. aB2GPI

Line 120. aPLA

Lines 123-124. Please use abbreviated form for theses autoantibodies

Symptoms, please correct

How was lupus nephritis diagnosed? Please add information about classes of LN.

Line 188. Interestingly, what kinds of vasculitis were overlapping with SLE?

Tables in the manuscript and in additional materials differ.

What is the purpose of the beginning of the discussion section regarding main aims of the study?!

et al.

Some parts of the conclusions does not reflect the main results. Please highlight the clinical-relevant information for practitioners.

Comments on the Quality of English Language

Moderate editing of English language required.

lupus antikoagulant

Problems with typos, English grammar.

Author Response

Thank you very much for your extensive work.

Reviewer 2 Report

Comments and Suggestions for Authors

The authors answered my concerns. However, their explanations are general and more appropriate for a review than an original article. I would like them to concretely show how their findings change the management of the patients from their sample (with numbers), in the Discussion section.

Author Response

Thank you very much for your helpful comments
